# Molecular Characterization and Phylogenetic Analysis of Lumpy Skin Disease Virus Collected from Outbreaks in Northern Thailand in 2021

**DOI:** 10.3390/vetsci9040194

**Published:** 2022-04-18

**Authors:** Tawatchai Singhla, Kittikorn Boonsri, Khwanchai Kreausukon, Wittawat Modethed, Kidsadagon Pringproa, Nattawooti Sthitmatee, Veerasak Punyapornwithaya, Paramintra Vinitchaikul

**Affiliations:** 1Ruminant Clinic, Department of Food Animal Clinics, Faculty of Veterinary Medicine, Chiang Mai University, Chiang Mai 50100, Thailand; tawatchai.singh@cmu.ac.th (T.S.); khwanchai.kreau@cmu.ac.th (K.K.); veerasak.p@cmu.ac.th (V.P.); 2Center of Excellence in Veterinary Public Health, Faculty of Veterinary Medicine, Chiang Mai University, Chiang Mai 50100, Thailand; 3The Veterinary Diagnostic Center, Faculty of Veterinary Medicine, Chiang Mai University Animal Hospital, Chiang Mai 50100, Thailand; kittikorn.boosri@cmu.ac.th (K.B.); kidsadagorn.p@cmu.ac.th (K.P.); 4Herd Health Unit, The Fifth Regional Livestock Office, Department of Livestock Development, Ministry of Agriculture and Cooperative, Muang, Chiang Mai 50300, Thailand; wittawatm@dld.go.th; 5Laboratory of Veterinary Vaccine and Biological Products, Faculty of Veterinary Medicine, Chiang Mai University, Chiang Mai 50100, Thailand; nattawooti.s@cmu.ac.th; 6Excellent Center in Veterinary Bioscience, Chiang Mai University, Chiang Mai 50100, Thailand

**Keywords:** lumpy skin disease, molecular epidemiology, GPCR, phylogenetic, Thailand

## Abstract

Understanding molecular epidemiology is essential for the improvement of lumpy skin disease (LSD) eradication and control strategies. The objective of this study was to perform a molecular characterization and phylogenetic analysis of lumpy skin disease virus (LSDV) isolated from dairy cows presenting LSD-like clinical signs in northern Thailand. The skin nodules were collected from 26 LSD-suspected cows involved in six outbreaks during the period from July to September of 2021. LSDVs were confirmed from clinical samples using the polymerase chain reaction (PCR). The PCR-positive samples were subsequently amplified and sequenced using a G-protein-coupled chemokine receptor (GPCR) gene for molecular characterization and phylogenetic analyses. All 26 samples were positive for LSDV by PCR. A phylogenetic analysis indicated that the 24 LSDV isolates obtained from cattle in northern Thailand were closely related to other LSDV sequences acquired from Asia (China, Hong Kong, and Vietnam). On the other hand, two LSDV isolates of the cows presenting LSD-like clinical signs after vaccination were clustered along with LSDV Neethling-derived vaccines. The outcomes of this research will be beneficial in developing effective control strategies for LSDV.

## 1. Introduction

The lumpy skin disease virus (LSDV) belongs to the genus *Capripoxvirus* in the family *Poxviridae* and is known to cause lumpy skin disease (LSD) [1]. LSD is characterized by nodular skin lesions, fever, and enlargement of the lymph nodes. This disease impacts not only cattle but also the economies of affected countries across the world. These economic impacts are due to losses associated with animal production, restrictions in the trade of live animals and their products, and the costs of vaccination and treatment programs [2]. It has been estimated that the economic losses in the dairy business are 141 USD per lactating cow, while the diagnostic and treatment costs have been estimated at 5 USD per cow [3].

LSD was discovered in Zambia in 1929 and then spread to most areas of Africa. Consequently, the disease gradually spread to the Mediterranean basin, Europe, Middle East, and Asia [4,5]. In 2019, LSDV was reported in Bangladesh, China, and India. Subsequently, the disease was discovered in Nepal, Sri Lanka, Vietnam, Myanmar, Thailand, Malaysia, Laos, and Cambodia [6,7,8]. Phylogenetic analysis by sequencing of the G-protein-coupled chemokine receptor (GPCR) gene demonstrated that LSDVs circulating in Asian countries are the same and have a close genetic relationship with the Russian LSDV strain [9]. Conversely, it has been reported that LSDV in India was genetically close to the South African strains [10].

Eradication and control measures for LSD are comprised of cattle movement restrictions, the dissemination of vaccinations, and disinfection and arthropod control protocols [8]. A live-attenuated LSDV vaccine is a useful tool for LSD control worldwide. However, biosafety, efficacy, and transmission capacity of the live-attenuated LSDV vaccine should be considered [2,5]. Adverse effects of LSDV post-vaccination have been reported in animals. These include the occurrence of LSD-like skin nodules and fever [11]. Additionally, researchers in Russia have identified recombinant a vaccine-like LSDV that was isolated from infected animals. This finding was determined to be a new LSD outbreak rather than a continuation of the field-type epidemic [12]. However, many researchers are still unclear on the mechanism of recombination, which would need to be further investigated.

Understanding molecular epidemiology is necessary for the improvement of LSD eradication and control strategies [13]. Genetic characterization and phylogenetic analysis of LSDV during outbreaks provide critical LSDV data such as the level of transboundary circulation, disease hotspot areas, and the origins of LSDV [13]. Sequencing of the GPCR gene has been used by various researchers for the purposes of characterization or in molecular epidemiological studies involving LSDV [10,12,13,14]. A study in Uganda employing GPCR gene sequencing has determined that these LSDV sequences are closely related to sequences obtained from neighboring East African countries and recent outbreaks in Europe. This critical information is essential in better understanding LSDV molecular epidemiology. Furthermore, a knowledge of molecular epidemiology can support policy-makers to improve LSD eradication and control measures of the country [13].

Thailand is predominantly an agricultural country with a cattle population of approximately 10 million heads. Most of farms are small-holders with a herd size of 20–30 animals per farm. Animals are kept in open-yard housing with semi-intensive managements. The cattle are mostly raised for domestic consumption. However, it has been reported that about 100,000 heads of large ruminants were exported to neighboring countries [15]. The first outbreak of LSD was reported in the northeastern region of Thailand in March 2021. Thereafter, the disease has spread throughout the country and severely affected a lot of cattle [16]. The northern part of the country shares borders with a number of neighboring countries. It has been assessed that Thailand is facing a high risk of LSD outbreak due to its informal live animal importation from neighboring countries where the disease has already been introduced [15]. Since the first outbreak in Thailand, the Thai government has launched LSD control measures including the restriction of cattle movement to within 50 km radius of the outbreak area. Furthermore, live-attenuated Neethling LSD vaccines have been disseminated for disease control throughout the country. However, 60% of animals were vaccinated, and LSD-like clinical signs have been observed in the north and around the country after the control measure was applied. However, there is no available data on the molecular characterization and phylogenetic analysis of the circulating virus that had been isolated from infected and vaccinated animals in the northern part of the country. Therefore, this study aimed to perform studies on the molecular characterization and phylogenetic analysis of LSDV with use of the GPCR gene isolated from cattle presenting LSD-like clinical signs in the northern part of Thailand.

## 2. Materials and Methods

### 2.1. Study Area and Sample Collection Process

This study was undertaken in four districts of Chiang Mai Province, including Mae Wang, Doi Lo, Mae On, and San Pa Tong Districts, and two districts of Lamphun Province, namely Ban Thi and Ban Hong Districts (Figure 1). Skin nodules were collected with a diameter of 6 mm using Biopsy Punches (SMI, Sankt Vith, Belgium) following disinfection of the biopsy area from 26 LSD-suspected cows that were involved in six outbreaks during the period of July–September 2021. Out of 26 clinical samples, 24 samples were collected from cows presenting LSD-like clinical signs that had previously not been vaccinated against LSDV. The other two samples were collected from cows presenting LSD-like clinical signs after receiving the live attenuated vaccine (LUMPYVAC^®^; Vetal Animal Health Product SA; Adiyaman; Turkey). Furthermore, 1 sample of a commercial vaccine which was used in those farms was also collected for the PCR and the GPCR gene sequencing of a live-attenuated Neethling virus to compare its genetic characteristics with LSDVs isolated from the clinical samples. Clinical samples including skin biopsies and scabs were collected in sterile cryovials using an aseptic technique, as has been described by the OIE [17]. Additionally, more in-depth information of suspected LSD-infected cattle such as age, gender, breed, and clinical signs was also collected. After collection, each clinical sample was labeled by a unique sample ID. The samples were kept in a cooler box with ice packs until transferring to the Veterinary Diagnostic Center, Faculty of Veterinary Medicine, Chiang Mai University. The samples were then stored at −80 °C for further molecular analysis.

### 2.2. DNA Extraction

Clinical samples of skin biopsies and scabs were sliced into small pieces at a weight of about 400 mg using a scalpel and homogenized in 500 μL of sterile 1X PBS solution at a pH of 7.4. Then, the tissue homogenates were processed for DNA extraction using a NucleoSpin^®^ Tissue kit (Macherey-Nagel, Duren, Germany), as described by the manufacturer’s instructions.

### 2.3. Lumpy Skin Disease Virus Confirmation

The presence of *Capripoxvirus* was confirmed with the use of PCR by amplification of a 192 bp region in the P32 gene using a pair of primers: forward primer, 5′-TTTCCTGATTTTTCTTACTAT-3′, and reverse primer, 5′-AAATTATATACGTAAATAAC-3′, according to the PCR procedures, as described in a previous study [18]. The PCR reactions were prepared in a final volume of 50 μL mixtures containing 25 μL of Quick Taq^TM^ HS DyeMix (Toyobo, Japan), 21 μL of PCR water, 2.0 μL of extracted DNA, and 1.0 μL of each 10 μM primer concentration. Then, the PCR was constructed using a Bio-Rad S1000 ThermoCycler (Bio-Rad, Watford, UK) under thermal conditions, including an initial denaturation at 94 °C for 5 min followed by 34 cycles of denaturation at 94 °C for 1 min, primer annealing at 50 °C for 30 s, a standard extension at 72 °C for 1 min, and a final extension step at 72 °C for 5 min. After that, the PCR products were separated in 1.5% Agarose gel and visualized under UV transilluminator for confirmation of LSDV-positive samples with a band size of 192 bp.

### 2.4. GPCR Gene Amplification

All positive samples were processed to amplify the GPCR gene using PCR. PCR was performed using primers with the appropriate sequences (5′-TTAAGTAAAG CATAACTCCAACAAAAATG-3′ and 5′-TTTTTTTATTTTTTATCCAATGCTAATACT-3′) which were created for amplification of the entire GPCR gene at position 6961–8119 in the LSDV genome [19,20]. Two additional primers, according to Le Goff et al. (5′-GATGAGTATTGATAGATACCTAGCTGTAGTT-3′ and 5′-TGAGACAATCCAAACCACCAT-3′), were used for further DNA sequencing as previously described [20]. To amplify the GPCR gene, the PCR reactions were prepared in a reaction volume of 50 μL containing 25 μL of Quick Taq^TM^ HS DyeMix (Toyobo, Japan), 21 μL of PCR water, 2.0 μL of the extracted DNA, and 1.0 μL of each 10 μM primer concentration. Then, the PCR was constructed under the following thermal conditions, including an initial denaturation step at 96 °C for 5 min followed by 35 cycles of a final denaturation at 95 °C for 30 s, an annealing at 50 °C for 30 s, an extension step at 72 °C for 30 s, and a final extension step at 72 °C for 5 min. After that, the PCR products were resolved on 1.5% agarose gel against a SibEnzyme^TM^ 100 bp DNA ladder (SibEnzyme^®^, Nowosibirsk, Russia) at 100 V in 1X Tris-Acetic acid-EDTA (TAE) buffer containing 0.5 μg/mL RedSafe^TM^ (iNtRON Biotechnology, Sangdaewon-dong, Korea) for 30 min.

### 2.5. Nucleotide Sequencing and Analysis

Following agarose gel electrophoresis on 1.5% agarose gel, the PCR amplicons were identified as the expected size using a molecular-weight marker. Then, the selected DNA bands were trimmed and purified using gel purification (Macherey-Nagel, Duren, Germany) according to the manufacturer’s instructions and transferred to Bio Basic (Ontario, Canada) for DNA sequencing. After that, the quality of the obtained sequences was checked, and the ends of the sequences were trimmed using Molecular Evolutionary Genetics Analysis (MEGA) version X software (Pennsylvania, USA). The similarity of the trimmed sequences was then tested with other GPCR gene sequences of LSDVs in GenBank using the National Center for Biotechnological Information’s (NCBI) web-based Basic Local Alignment Search Tool (BLAST). LSDV-specific signatures of these nucleotide sequences were also checked by translating the sequences to amino acid sequences followed by multiple sequence alignment using MUSCLE on the EMBL-EBI web server. After that, phylogenetic analysis was performed using MEGA X. LSDV sequences were selected after BLAST based on nucleotide similarity. Consequently, the origins of the isolates were representative of sequences from Asia, Europe, and Africa. Sequences were also selected from established LSDV vaccine strains, as well as those from the goatpox and the sheeppox viruses. Then, a phylogenetic tree was created using the Neighbor-Joining method in MEGA X, which used the evolutionary distances derived from identical branch lengths in the same sites to generate the phylogenetic tree [21]. The maximum likelihood method was used for the calculation of the evolutionary distance. The GPCR sequences of the northern Thai LSDVs were submitted to the GenBank database and are available under accession numbers ON024907 to ON024933.

## 3. Results

### 3.1. LSD Outbreak Investigation

Six suspected LSD outbreak areas in six districts of Chiang Mai and Lamphun Provinces were investigated in 2021. Suspected LSD cattle presented a range of clinical signs including depression, loss of appetite, fever, nasal and ocular discharges, enlarged superficial lymph nodes, circumscribed skin nodules on all parts of the body, and a decrease in body condition score (Figure 2).

All of the farms in this study were dairy farms and included eight farms located in Chiang Mai Province (one farm from Doi Lo District, three farms from Mae On District, two farms from Mae Wang District, and two farms from San Pa Tong District) and four farms located in Lamphun Province (one farm from Ban Thi District and three farms from Ban Hong District).

### 3.2. Lumpy Skin Disease Virus Confirmation

LSDVs were isolated from 26 dairy cows and 1 commercial vaccine were confirmed by PCR. Out of 26 dairy cattle, eight cows were dairy calves, with ages ranging from one to six months old. The rest included 17 lactating cows and one dry cow.

### 3.3. Phylogenetic Analysis

Phylogenetic analysis clustered the northern Thai LSDV isolates, vaccine strains, and goatpox and sheeppox viruses into separate clades within the *Capripoxvirus* family (Figure 3). There were three subgroups of the LSDV on the GPCR gene tree. Out of 26, the 24 LSDVs isolated from the LSD outbreaks in northern Thailand were closely related to other LSDV sequences acquired from Asia (China, Hong Kong, and Vietnam). On the other hand, two LSDV isolates of cattle presenting with LSD-like clinical signs after receiving vaccinations and one LSDV isolate of the commercial vaccine were clustered along with LSDV Neethling-derived vaccines. Multiple sequence alignments of the GPCR gene demonstrated that 22 isolates of the northern Thai LSDVs presented the 12-nucleotide insertion, as shown in Figure 4. However, four LSDV isolates (two isolates from infected herds and two isolates from cattle presenting LSD-like clinical signs after being vaccinated) had a different alignment at the same region as same as the isolate from the commercial vaccine. The LSDVs isolated from the northern Thai outbreaks revealed nucleotide sequence identities between 98 and 100% when compared with the other LSDV strains (e.g., LSDV/KM/Taiwan/2020, Dinh-To/Vietnam/2020, LSDV/HongKong/2020, and China/GD01/2020) obtained from GenBank. Likewise, the amino acid identities of the northern Thai LSDVs also ranged between 98 and 100% after being compared with other protein sequences of LSDVs available in GenBank (e.g., LSDV/Xinjiang/China/2019, Neethling2490/Kenya/1958, Khvalynsky/Russia/2018, and LSDV/IND/ODI/30PR-LT/India/2019).

## 4. Discussion

The present study investigated and confirmed LSD cases in northern Thailand using PCR. Additionally, LSDVs isolated from the identified outbreaks were characterized using GPCR gene sequencing. To the author’s knowledge, this is the first report on the molecular characterization and phylogenetic analysis of the LSDV that has circulated in this region in 2021.

The first LSD outbreak in Thailand was discovered in the northeastern part of the country, in March of 2021. Subsequently, the disease was reported in the northern part of the country to include Chiang Mai and Lamphun Provinces in May of 2021. LSD-suspected cattle presenting LSD-like clinical signs were examined, and tissue samples were collected for LSD-infection confirmation using PCR. The results of PCR confirmed that LSD outbreaks occurred in this region. Chiang Mai and Lamphun Provinces are home to the highest density of dairy cattle when compared to other provinces in northern Thailand. Moreover, Chiang Mai Province has borders with a number of neighboring countries, and the incidences of cattle movement across these borders have been reported. These factors may present an increased risk for LSD outbreaks in this region [15]. However, the links that exist among LSDVs circulating in the northern part of the country, the eastern part of Thailand, and neighboring countries would need to be further investigated.

After GPCR gene sequencing, out of the 26 LSDV isolates, 24 isolates were grouped with LSDV isolates acquired from China, Hong Kong, and Vietnam. This finding suggests that the same LSDVs play an important role for outbreaks across shared borders. It has been reported that Thailand has a high risk of LSD outbreaks because live animal trade across borders from neighboring countries is quite high [15]. Moreover, the animal trade industry is known to ship animals for thousands of kilometers and across several countries. For example, live beef cattle or buffaloes from India and Bangladesh have been transported by trucks and ships to their final destination in Vietnam or China by transiting through Myanmar, Thailand, Cambodia, and Laos [15]. This circumstance supports the suggestion of a link between the LSDV circulating in Thailand and the LSDVs existing in China, Hong Kong, and Vietnam.

Out of 26 LSDV isolates, two isolates were grouped in vaccine strains after phylogenetic analysis. These LSDVs were isolated from cattle presenting LSD-like clinical signs within 7–14 days after vaccination. Remarkably, the cattle were healthy before vaccination and were acquired from LSD-free herds before outbreaks had occurred in northern Thailand. This finding suggests that the clinical signs of these cattle, such as circumscribed nodule skins, may have resulted as adverse effects of the LSD vaccine. A study in Jordan reported that certain adverse effects of the LSD vaccine occurred in animals within 1–20 days after being vaccinated. These effects included fever, nasal and ocular discharges, and variable-sized cutaneous nodules [11]. However, the recombination of vaccine strains of LSDV with field strains should be monitored, since a recombinant vaccine-like LSDV was reported in Russia [12]. Thus, a further study needs to be done with a complete genome sequence analysis of the LSDVs isolated from northern Thailand for a better understanding.

The LSDVs circulating in northern Thailand were found to contain the 12-nucleotide insertion in the GPCR gene. This result is similar to that of a study conducted in India and Bangladesh, which also found the 12-nucleotide insertion of the GPCR gene in LSDVs isolated from outbreaks in those countries [14,22]. Furthermore, there have been reports that this insertion was also found in other LSDVs (LSDV KSGP 0240 and LSDV Neethling), namely the two historical field isolates acquired from Kenya (collected before 1960), the recombinant LSDVs acquired from Russia, and the recent LSDV isolates obtained from China [12,14]. Moreover, it has been reported that LSDVs isolated from all the new LSD outbreaks taking place in Asia carried the 12-nucleotide insertion in their GPCR gene [23]. Importantly, the existence of the 12-nucleotide insertion indicates that the LSDV originating from northern Thailand is different from the LSDVs circulating in Africa, Europe, and the Middle East [14,20,24,25]. However, a further study with a complete genome sequence is needed to clarify the difference of the partial sequence alignment between the 22 isolates and 4 isolates and to monitor a gene mutation of the LSDVs.

## 5. Conclusions

This study firstly provides information on the molecular characterization and phylogenetic analysis of LSDV circulating in northern Thailand using GPCR gene sequencing. The results of this study indicate that the LSDV isolated from this region was genetically similar to the LSDVs circulating in China, Hong Kong, and Vietnam. This finding strongly supports the transboundary spreading of LSDVs throughout Asian countries. This information will be beneficial to researchers in better understanding LSDV molecular epidemiology and will contribute to the existing efforts of the government of Thailand in developing effective control strategies for LSDV.

## Figures and Tables

**Figure 1 vetsci-09-00194-f001:**
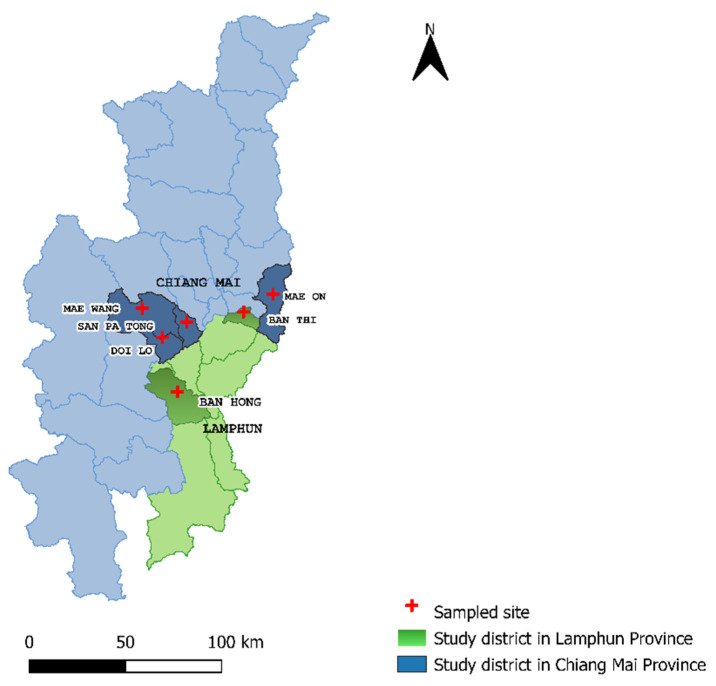
Location of the study areas. Sample sites are shown in dark blue and dark green for the districts of Chiang Mai and Lamphun Provinces, respectively.

**Figure 2 vetsci-09-00194-f002:**
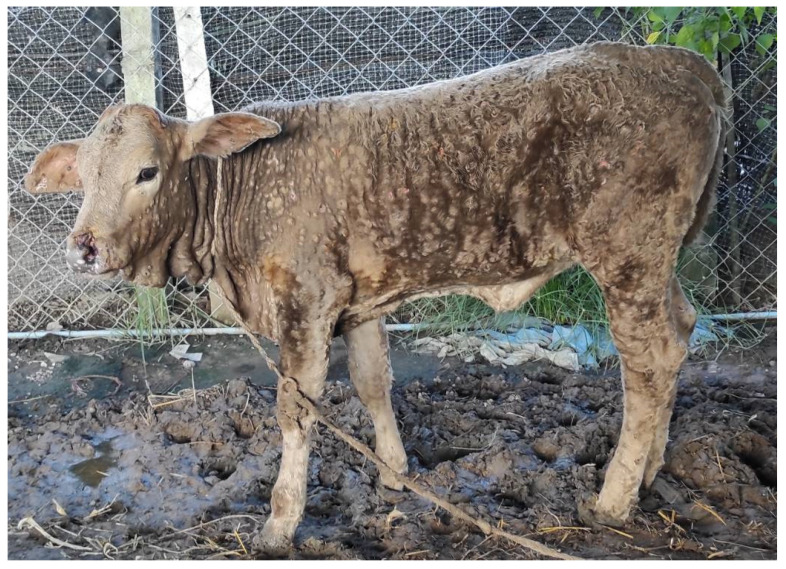
Dairy calf was infected with lumpy skin disease virus and presented a number of circumscribed skin nodules on the entire body.

**Figure 3 vetsci-09-00194-f003:**
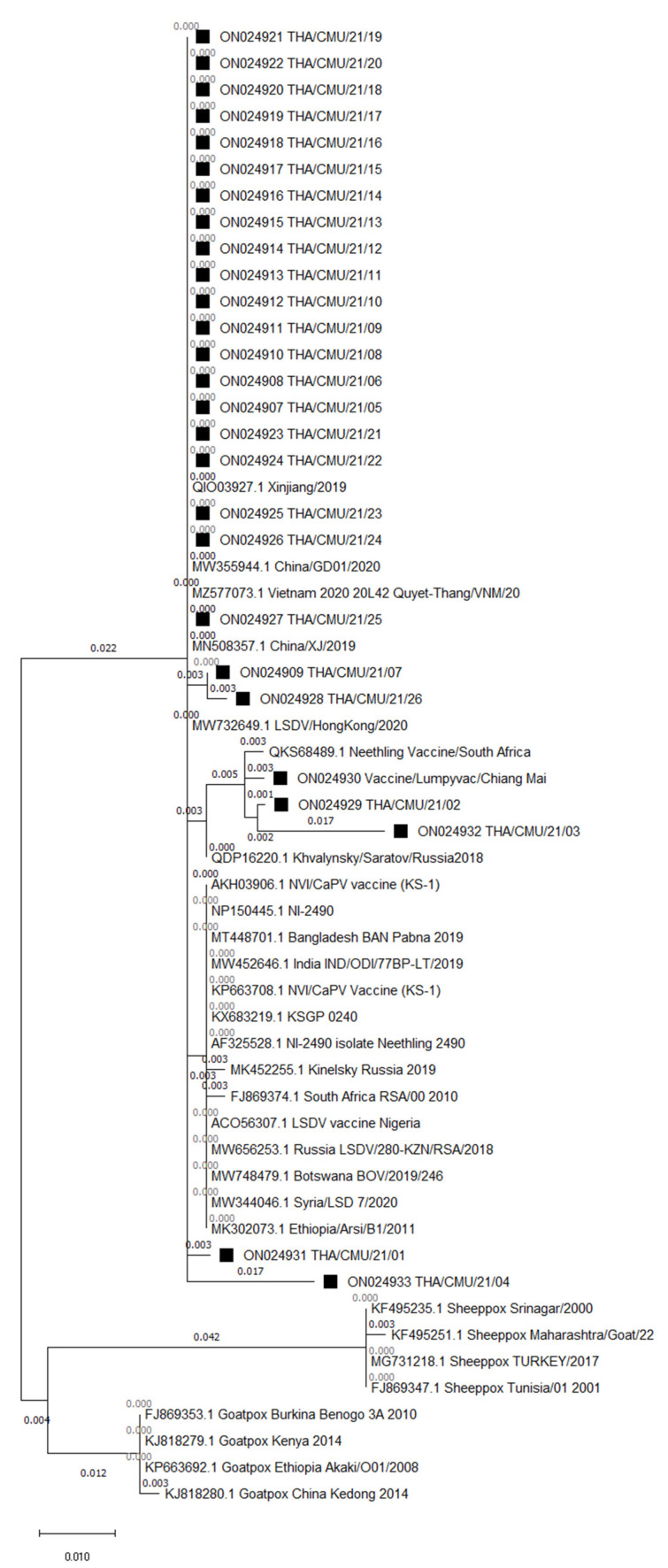
Phylogenetic tree demonstrating the relationship between GPCR gene sequences obtained from northern Thailand (marked with black squares) with other *Capripoxvirus* GPCR gene sequences acquired from GenBank.

**Figure 4 vetsci-09-00194-f004:**
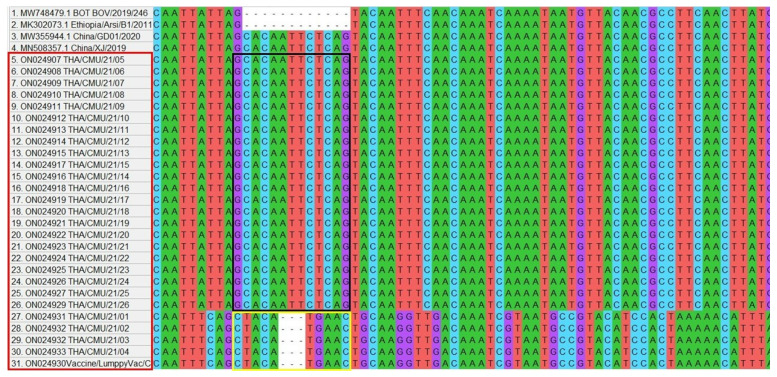
Multiple sequence alignment of GPCR sequences of 22 LSDV isolates obtained from northern Thailand presented 12-nucleotide insertion, the same as in the China strains. Four LSDV isolates had a different alignment at the same region (marking in a black and yellow horizontal rectangular shape, respectively).

## Data Availability

Data available on request due to privacy for sequential projects.

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
