# Peer review of "Molecular Characterization and Phylogenetic Analysis of Lumpy Skin Disease Virus Collected from Outbreaks in Northern Thailand in 2021"

_vetsci, 2022, doi:10.3390/vetsci9040194_

Round 1

Reviewer 1 Report

Summary

The submitted manuscript by Singhla et al., describes lumpy skin disease viruses isolated in Thailand based on the GPCR gene. Skin biopsy samples were used and virus isolated from these was sent for Sanger sequencing. Sequencing revealed that  the sequences from Thailand clustered with most Asian strains. The authors hypothesize the role of animal movement accross boarders as a likely route of spred.

In text specific comments.

Double full stops line 59 and 65

Line 82 – which vaccine?

Skin nodules were collected (line 95) or biopsies from nodules, there is need to be specific in the methods section….e.g. an 8 mm biopsy was collected  from the nodule site.

Figure 1 and the meaning of the light shaded areas?

Figure 3 - legend to be improved, what the accession numbers are for.

Line 136 – what does placed internally means? Sentence can be improved for quality.

Line 231 - in the vaccine strain seems inappropriate, please revise the grammar there.

232-238, What did analysis for animals that had been vaccinated reveal? Did you study these and if so were they vaccine strains? Is there a possibility of vaccination of infected calves?

There is not enough understanding of LSDV vaccine strains. The viruses stated in line 244 are virulent WT.

Line 262 – have not has.

General comments

With the advent of NGS, the report strikes me more as a communications type paper as more information from the extracted DNA or the samples could or should be added. Also, it is stated in the paper that translated amino acid sequences were made but I did not understand if the tree was made using them or nucleotides and if so, could you merge the two as it appears in Chibbsa et al., 2021. Another idea could have been to include a lot more sequences to make the tree more informative.  Also, you could sequence another conserved gene to cement you findings.

Reviewer 2 Report

vetsci-1628502-peer-review-v1

The article is well written and for the most part easy to follow and to understand. So well done to the author(s). Only the section regarding the phylogenetic tree should be clarified more and some additional sequences should be added to the analysis.

References to text are made between quotes “….”

Page 2 line 46

“LSD was discovered in Zambia in 1929 and subsequently spread to many European and Asian countries”

>>As it is written know, it is as LSDV jumped from Zambia to Europe and Asia. In reality it is a more gradual process with a northbound migration towards the Mediterranean basin and from there to Europe, Middle East and Asia.

Page 2 line 59 and line 65

“and fever [11]..”

“control strategies [13]..”

>> two terminal punctuations after [11] and after [13]

Page 3 line 119

“AAATTATATACG TAAATAAC”

>>Why is there a space in the primer sequence?

“The presence of the LSDV specific nucleic acid was confirmed “

>> Caution needs to be paid here. The primers listed are not LSDV specific  as they also amplify sheep and goatpox. These primers are capripox specific.  The fragment length can be different between the SPPV, GPV and LSDV but I doubt that they can be clearly separated on a 1.5% agarose gel. The fact that the PCR is positive does suggest that is LSDV as SPPV and GPV do not infect cattle. Caution needs to be paid if a heterologous vaccine is used and the sampling is done quickly after the vaccination.  I would change the sentence slightly by saying: “the presence of capripox vius was……”

Page 4 line 163

“Tamura 3 parameter model with 1000 bootstrap replications.”

>> Was there a specific reason for selecting the Tamura 3 model.

Page 5  Figure 2 legend line 175 to 177

This is repeating what is written in the text itself (169 to 173). Should be shortened.

Page 5 line 187 and following

This section is difficult to follow. Based upon the text above, there are 26 positive samples. However, I count only 13 black boxes in Figure 3. So not all were sequenced? Or not all were depicted in Figure 3. Why?  Or add some explanation why those were selected.

“Phylogenetic analysis was used to cluster the northern Thai LSDV isolates and vaccine strains into separate clades within the Capripoxvirus family (Figure 3). Goatpox and sheeppox viruses were also clustered in separate clades. There were three subgroups of 190 LSDV on the GPCR gene tree."

>> This is confusing to read. There are 2 clades of LSDV and three subgroups on the LSDV tree. Are these 3, the 2 LSDV clades + another? It is better to first discuss the overall structure of the tree and then zoom in.  Also indicate on the tree the 2 LSDV clades ! 

In the figure there is also a sample called Lumpyvac/Chiang Mai. Is this also a field sample as there is no accession number.

“On the other hand, two LSDV isolates of cattle presenting with LSD-like clinical signs after receiving vaccinations were clustered along with 194 LSDV Neethling derived vaccines.”

>> There is a branch with 3 black boxes. Are that the 2 field sample + the vaccine derived sequence. Have you obtained this sequence ? How? This is not clear.  Also there are several vaccine sequences in Genbank. Add at least one to the tree !

Page 6 figure 3

>> resolution should be improved

>> include a few more SPPV in the tree: at least the same number as GPV.

Overall remark

It would have been interesting to have some LSDV DIVA real-time PCR results on those samples as well.

Reviewer 3 Report

The authors report the identification and phylogenetic analysis of 26 LSDV positive skin samples collected in northern Thailand in 2021. For the Capripoxvirus identification a p32 based PCR assay was used and for the pylogenetic analyses the GPCR gene was amplified and sequenced.

The manuscript is mostly well written and deliver new results about the ongoing spreading of LSDV in Asia. Nevertheless, some information about the LSD situation in Thailand is missing. In addition, the generated sequence data and resulting alignments are not present very well. Furthermore, the languages need some minor corrections.

Major comments:

  • In the introduction, more information about the cattle production in Thailand should be included, in general (number of cattle, centre of cattle production, housing conditions, mean herd size, trade with neighbouring countries, …)
  • More information about the control measure in context with the LSDV outbreaks are welcome: When was the first outbreak clinically reported? Were restriction zones defined? Is the trade outside the restriction zones banned? What LSDV vaccine was used for control? What is the producer or supplier of the vaccine? How many cattle were vaccinated in Thailand with the live-attenuated vaccine (vaccination rate)?
  • The sequence data of the GPCR genes must be submitted to GenBank.
  • The accession numbers must be included in the manuscript as well as in the phylogenetic tree.
  • A novel figure with a partial sequence alignment (e.g. 50-100 nt) of the GPCR gene of all 26 strains from Thailand and further strains covering the 12-nucleotide insertion region should be prepared. Here, the LSDV strain defined in the discussion (Line 240 ff.) should be included. Based on this figure the characteristic 12-nt-insertion can be better understandable by the reader.
  • Please include the results of the BLASTN and BLASTP search for the novel GPCR sequences in the main text. Here, some of the most related strains (including country and year) should be defined.
  • In the discussion the limited character of the presented work regarding the detection of chimeric virus (like in Russia) should be noted. Alternatively, the analysis full length sequences of one or more isolates would improve the manuscript substantially. But perhaps, this could be a part for the next paper.

Minor comments:

  • Line 27-28: Remove the sentence or define the relevance of the 12-nt insertion more clearly. The reader cannot understand, why the 12-nt insertion is important.
  • L37+38: Capripoxvirus and Poxviridae should be in italics.
  • L46: The sentence suggested that LSDV was first detected in Zambia and switching than to Europe and Asia. Please correct it. LSDV is widely reported and spreading in Africa and at the end of 20th century and beginning of 21st century the virus is switching to several countries in the Far East, Asia and Europe.
  • Line 59+65: remove the additional dot.
  • Line 65-67: What you mean with the sentence. Please define more clearly.
  • Line 65+85: an additional space at the beginning of the sentence is necessary.
  • Line 228: Vietnam

Round 2

Reviewer 1 Report

Considerable changes have been made and are satisfactory. The authors need to conform with what other authors do in terms of referencing sources of vaccines and punch biopsies. General comment on English grammar and word flow, kindly, seek assistance from an editing service. A good example of where this is needed is in the introduction. In line 80, you talk about LSD severely affecting Thailand,  then next sentence you talk about the first case (line 82 a beef cattle is not good grammar), and you end with subsequent spread. Ideal flow would state that LSD was detected in March...2021 and has spread...and ultimately is severely affecting a lot of cattle.

Line 74-75 needs to be reworded otherwise, it looks similar to a statement in Ochwo et al., 2021. 

Reviewer 3 Report

The modification and added information in the revised manuscript are fine for me.
